# Improvement of Container Terminal Productivity with Knowledge about Future Transport Modes: A Theoretical Agent-Based Modelling Approach

**Mehdi Mazloumi [1] and Edwin van Hassel [2,*]**

1   C-MAT, Center for Maritime and Air Transportation Management, Antwerp Management School, University of Antwerp, Boogkeers 5, 2000 Antwerp, Belgium; Mehdimazloumi6@gmail.com
2   Department of Transport and Regional Economics, University of Antwerp, Prinsstraat 13, 2000 Antwerp, Belgium
*   Correspondence: Edwin.vanhassel@uantwerp.be; Tel.: +32-(0)3-265-4111

**Abstract:** Despite all the achievements in improving container terminal performance in terms of equipment and container stacking systems (CSS), terminal operators are still facing several challenges. One of these challenges is the lack of information about further transportation modes of the container, which leads to extra movements of the container inside the stacking area. Hence, we aimed to examine factors that affect container handling processes and to evaluate a container terminal's overall equipment effectiveness. This study used data from a container terminal at the Port of Antwerp, Belgium. An agent-based model was developed based on a block-stacking strategy to investigate two scenarios: (1) having information about further transportation modes and (2) a base scenario. The Overall Equipment Effectiveness Index (OEE) was also adopted to evaluate the container terminal's effectiveness in both scenarios. Results showed that having information on further transportation mode significantly increased the container outflow, and the OEE index improved compared to the base scenario's results. Therefore, we recommend an integrated data-sharing system where all the stakeholders can share their information with no fear of losing their market share.

**Keywords:** container transportation; container stacking strategy; agent-based model; overall equipment effectiveness

## 1. Introduction

Due to a sharp rise in international maritime shipping, interest in container terminal operations for the seaside and landside has increased [1]. Transportation is a derived demand which connects different stakeholders locally, regionally, and internationally [2]. Therefore, to transship a container from an origin to a destination (i.e., the end-user), a collaboration among these stakeholders is needed [3]. These processes can be divided into three stages: (1) inbound containers arrive by vessels or barges and (2) are then transported to a container stacking area, where (3) containers are temporarily stored for later transportation either by trains, trucks, vessels, and barges. During all these stages, terminal operators need information about containers' further transportation modes to set up their loading/unloading planning and determine the number of pieces of equipment needed for the container handling process [4].

Meanwhile, container ship size has increased in recent years due to shipbuilding technology and the growing economy, leading to more complicated loading and unloading plans. The inefficient transfer of containers from ship to loading area is an important problem faced in container-handling facilities [5].

In total, 3928 million twenty equivalent unit (TEUs) were carried out on the northwestern European container ports in 2019. This is an increase of 34% compared to the volumes in 2010, emphasizing the importance of container transportation for the international market [6].

An increase in the share of maritime transportation has created a dynamic interface for hinterland transportation, as it has helped to evolve world trade [7] through so-called economies of scales, the impact of container ship sizes on maritime transportation, and the reduction of transportation costs [8]. Having considered all this progress in global trade, maritime transportation improvement depends on setting up an interorganizational information exchange connection among different organizational users, including manufacturers, customers, shipping lines, and customs and port authorities from different countries [9].

Container transportation has so far contributed to the development and performance of container terminal activities. To adopt this trend, not only container terminal operators but also supply chain operators should be able to improve the efficiency of port logistics. These objectives are achievable through developing a control algorithm and scheduling [10–12]. More so, terminal equipment efficiency, automation, and data integration as one of the key operational bottlenecks were widely used to improve container terminal productivity [13,14].

In container transportation, having information about transportation modes is essential among the terminal operators, the shipping companies, and the freight forwarders. Oftentimes, there is no clear information on how a container proceeds, and this leads to delay in containers transportation flow. This could disrupt the container terminal's scheduled stacking plan, which creates uncertainty about its stacking plan, high waiting times for container transportation from an origin to a destination, and occupied container terminal stacking capacity. These problems have led to a varied discussion among the various parties involved in container transportation; several different software programs and other solutions have been proposed to improve the situation, yet there has been no final answer.

In order to improve container terminal efficiency, it is of particular importance to consider the harmony of different container-handling equipment. In a container terminal, various cargo-handling equipment is used, including quay cranes (QCs), container transportation vehicles (CTVs), straddle carriers (SCs), and yard cranes (YCs).

In a container terminal, container transportation flows are affected by several unpredictable challenges. A dynamic unloading/loading planning procedure should be considered to minimize SCs queuing or disarranging and to maximize the container terminal's overall equipment effectiveness. Some of the challenges on the container terminals are:

- Uncertainty of ships' exact arrival time (in general, the planners of a container terminal can plan QCs/YCs worklist only a few hours prior to vessel arrivals) [15];
- Container terminal operators are unable to process further container transport modes (they don't know if the containers are going further with another deep-sea vessel, barge, rail, or road transport) [16];
- Most importantly, if they use more SCs at a peak time, there will be congestion of SCs [17].

In container transportation literature, the adverse effects of demand uncertainty or troubles in services are commonly acknowledged [18–20]. Specifically for container terminal stacking planning, the low reliability of deep-sea vessels, having no information about further transportation modes, and equipment preparation are major issues in operational level planning. In addition to the uncertainty in container arrivals, dynamism complicates container terminal planning. A terminal operator has to decide which QCs/YCs to load/unload containers, and which stacking areas are dedicated to inbound/outbound containers. During this time gap, other information impacts the terminal operator's decision making, such as new coming orders, delays, and cancellations [21]. Uncertainty and dynamism lead to a situation in which the efficiency of operational plans and the competitiveness of container terminals decrease.

Considering all these issues, the main goals of this paper are to examine factors that affect the container handling process and to evaluate a container terminal's overall equipment effectiveness.

More efficient container handling between the quay and yard sides minimizes cargo handling time, operational costs, and negative environmental effects. More importantly, container terminals gain an excessive loading/unloading operational capacity. Otherwise, they have to enhance their existing infrastructure (e.g., buying new QCs, YCs, and SCs). Various stakeholders, including shipping lines, freight forwarders, and terminal operators, would benefit from efficient container handling. Thus, this study seeks to find answers to the following questions:

1. What is the current stacking strategy at the container terminal of the Port of Antwerp?
2. How does information about further transportation modes impact the container transportation outflow and its overall equipment effectiveness?

In answering the research questions, we structured the rest of paper as follows: Section 2 concentrates on the literature review and previous works in this domain. The methodology and research approach are discussed in Section 3, while the attained results are shown in Section 4. The study's findings are discussed in Section 5. Finally, the related conclusion of the research is drawn in Section 6.

## 2. Literature Review

Maritime transportation, despite all technological improvements, mainly in the shipbuilding and engineering sectors, is still suffering in terms of port operation in loading/unloading procedures. Not much has changed over the past decades. On the other hand, containerized seaborne trade and container ship size has continually increased in recent years [22]. Container throughput has increased from around 200 million twenty-equivalent units (TEU) in 2000 to more than 829 million in 2020, which is evidence of the increasing maritime trade volume [23]. However, container transportation between the quay and yard sides is still the same as before in terms of data sharing among different stakeholders [20].

Given the fact that the loading and unloading process is an integrated process, the main part of this study has focused on the relationship between stacking strategies and the level of information about further transportation modes. A study conducted by Luo and Wu showed that the efficiency of transportation between the quayside and yard side might have a significant impact on the terminal's productivity [15]. Therefore, container terminal equipment should be operating as an integrated system to improve the terminal's performance. If SCs don't arrive on time, it will cause delay or congestion in QC/YC operations, ultimately decreasing effectiveness. If SCs arrive earlier, it might result in traffic congestion. Therefore, these problems impose various impacts, such as increased handling time, higher fuel cost, and higher environmental impact. Besides the importance of mobile equipment, fixed equipment (such as QCs) has significant importance in container terminal productivity. QCs are the most expensive equipment in the container terminal, with very high operational and capital costs. Therefore, the terminal operators tend to optimize QCs' performance as much as possible. YCs also affect container terminal operation and improve terminals' productivity [24].

Currently, three main container transportation vehicles can be distinguished based on the vehicles chosen: automated guided vehicles (AGVs), trucks, and straddle carriers (SCs). However, Gharehgozli et al. evaluated possible new layouts for container terminals in which traditional transportation methods may disappear completely, such as container racks, double story, ultra-high warehouse, super dock, robotic container management system (RCMS), and automated container transport system (ACTS) [25]. However, a major number of container terminals have automated a part of their container transportation system, but not made an absolute change to manage container movements inside the terminals. It seems that the abovementioned systems are too futuristic and may not be applied in container terminals too soon.

In order to evaluate the container terminal productivity, it is of particular importance to consider different port management approaches. Generally speaking, productivity is a measure of efficiency that can be attained from a certain ratio of outputs to inputs.

Nowadays, container terminals are competing to have a greater market share. In such a situation, productive container ports may be the winners of this never-ending competition. Compared to the public ones, private terminal operators are more efficient [26]. A green port approach that may help reach productivity goals effectively is to change both the behavior and performance of container terminals operations by improving effectiveness with green performance [27]. It can be concluded that a successful container terminal should be able to minimize its cost, and a good understanding of cost- and revenue-sharing schemes will help them improve the total profit of the port. Different methods have been used to analyze the terminal handling costs, such as the game theory model [28], Cournot's simultaneous quantity-setting [29], and Markov theory [30].

Regardless of port management approaches, the terminal operators should be able to manage their container stacking flow. The container stacking problem (CSP) is the main factor affecting the container terminal's unloading/loading process [31]. It refers to a problem that consists of determining containers' exact locations in a terminal stacking area. The CSP is a complex system, consisting of various dynamic and continuous interactions inside the stacking area and other elements, including ships, cranes, SCs, AGVs, or trucks [32]. To improve the overall terminal performance, several stacking strategies have been proposed. Different strategies in container terminals vary from terminal to terminal. Based on the results of Ma and Kim (2012), a stacking strategy can be divided into three main parts: block, bay, or stack [3]. Moreover, Rekik and Elkosantini (2019) categorized stacking rules based on Block Assignment Rules, Bay Assignment Rules, and Slot Assignment Rules (see Figure 1) [31]. Block assignment strategy deals with selecting the appropriate block for incoming/outgoing containers—for instance, a dedicated area for a specific type of container. The Bay Assignment strategy consists of allocating storage space for a specific group of containers and assigning containers of the same group to adequate bays [33]. The Slot Assignment strategy is defined for determining the exact storage location in the pre-selected bay (based on the Bay Assignment strategy) of the pre-selected block.

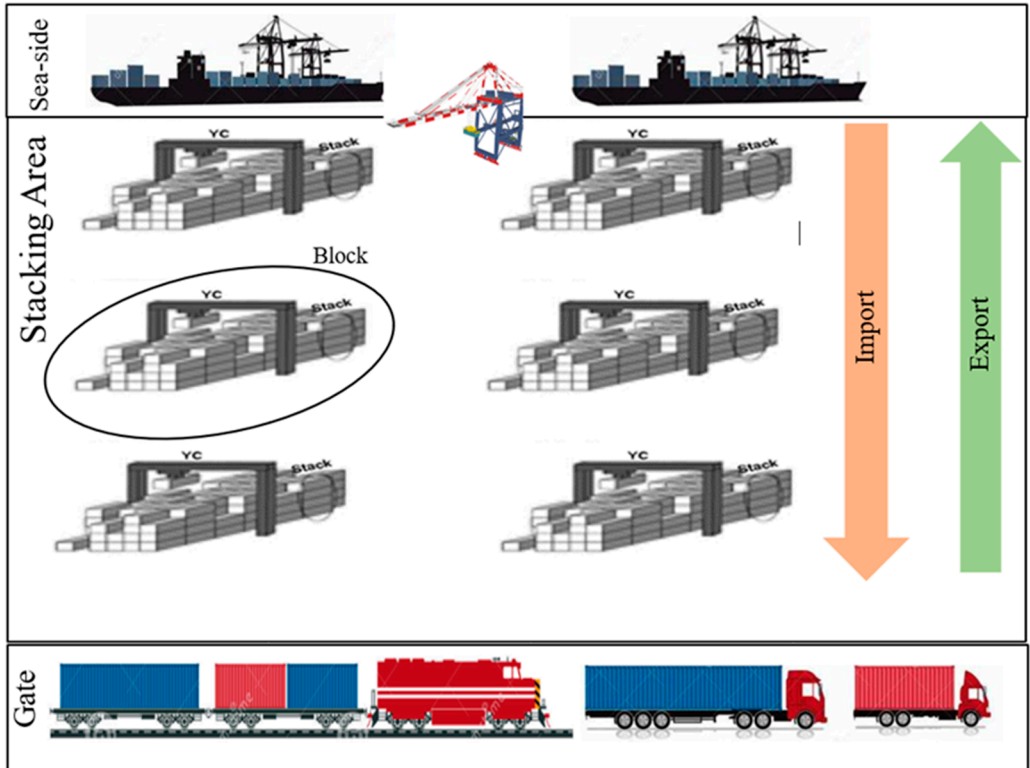

**Figure 1.** Container terminal stacking strategies Source: own figure based on authors' compilation.

Previous studies have dealt with container terminal efficiency and stacking strategies. A majority of them have focused on operational analysis [34], scheduling of container-handling equipment [15], and optimization of the container handling process considering equipment efficiency [35], while the present study tries to fill the gap between stacking strategies and overall equipment effectiveness by developing an agent-based model to address the issues of container handling flow. Therefore, this study proposes different stacking strategy scenarios in which, in the base case, we don't have information about how the containers are handled further. Then, in an alternative scenario, we check if and how the stacking strategy could be different if we did have this information.

## 3. Methodology

### 3.1. Research Approach

A quantitative research approach was employed to reach the final conclusion, based on three steps, which are briefly described as follows:

1. Development of scenarios based on the literature review for a possible solution to solve the loading/unloading problems based on different stacking strategies between the quay and yard side.
2. Development of agent-based modeling to analyze the different scenarios earlier developed.
3. Utilizing the Overall Equipment Effectiveness (OEE) index for the two cases to measure how well the operation is run compared to its ideal and full potential.

In order to develop the two cases, the container loading/unloading process based on the block-stacking strategy is considered. The two scenarios were developed based on evidence from the literature where three stacking strategies of block, bay, and slot assignment were discussed [31]. Considering the existing stacking strategy used in a container terminal of the Port of Antwerp, the block stacking strategy was selected to investigate the determined scenarios. A schematic overview of the developed scenarios has been presented in Figure 2 below.

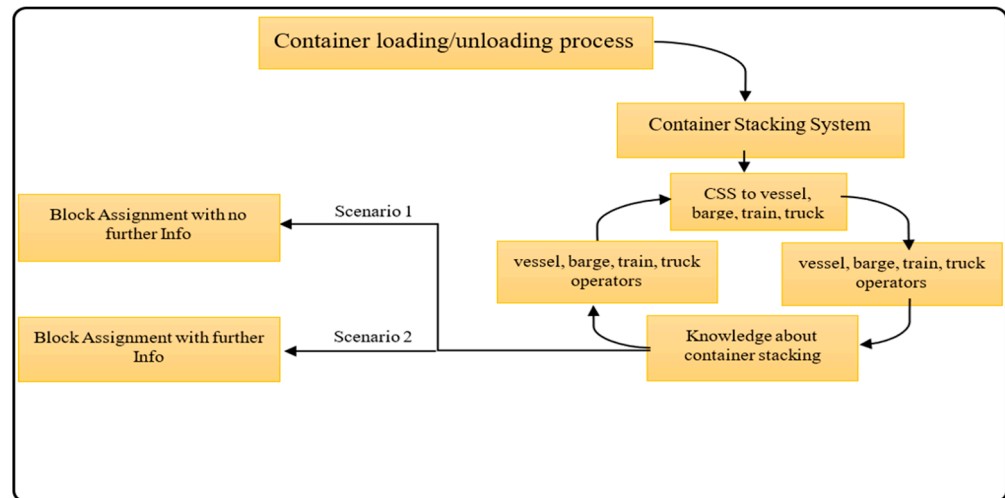

**Figure 2.** Container-handling scenarios.

Loading and unloading processes are shown by the loop in Figure 2. In order to unload a container onto a stacking area, the container stacking system (CSS) determines whether there is a dedicated space for a container or not. Then, the unloading process begins, considering knowledge about the container stacking area's status. However, in the loading process, containers' locations in the stacking area should first be identified, then vessel, barge, train, and truck operators should get information about the loading process.

The first scenario shows a situation of block assignment rules. In the base case, it means no information about how the containers should be handled further. The second

scenario examines a situation of block assignment rules in which where and how containers proceed further is known.

In this section, the Agent-Based Model (ABM) is explained, along with the developed Overall Equipment Effectiveness (OEE) index. Performance measurement is essential for all container terminals, especially in container terminal handling systems. A competitive advantage allows a container terminal to offer and sell services more attractive than those of its national and international competitors [36]. In this sense, OEE, as one of the commonly used performance indicators, was adopted for determining the container terminal's equipment utilization. Although this index initially appears linked to maintenance, it applies to broader activities to identify losses due to sustainability and establishes a complete understanding of the production process in terms of availability, performance, quality, and sustainability [37]. The different developed scenarios are explained in Section 3.4.

*3.2. Agent-Based Model*

Models and simulations are tools used to simplify existing complicated systems and allow the optimization procedure to be implemented before real-world model initialization [38]. The ABM was first developed by Uri Wilensky [39] to evaluate different scenarios using NetLogo. The ABM method has some advantages compared to traditional modeling approaches. Firstly, the ABM application has no agent selection limit, which basically provides the ability to evaluate each carrier across a set of variables. Secondly, ABM allows determining of the performance and the respective interactions using explicit modeling of behaviors and the interactions of each agent. Thirdly, in the ABM method, both agents and systems are able to memorize their actions in a dynamic modeling system [40]. The ABM is particularly well suited for complex systems over a time period. This makes it possible to find out the micro-level and macro-level patterns that emerge from agents' interactions [39]. Different models and techniques have been conducted to examine container-stacking strategies. A multi-agent approach was applied in a study conducted by Rekik and Elkosantini (2019) to minimize limitations related to online stacking strategies, distributed control, and efficiency [31]. The results of this study led to a system of container stacking with the ability to handle dangerous containers and decentralized control in an uncertain and disturbing environment. The inbound container volume, unloading, and stacking problems were evaluated using a two-stage search algorithm. Based on the formulation, an integer programming model is formed to decrease rehandling of containers and optimally distribute loading orders based on the stacking strategy [41].

Although several models have investigated the container unloading/loading problems with respect to stacking strategies, the main reason for the container handling problem with respect to the further mode of transport is not well documented. Furthermore, previous studies did not consider the main cause of the container handling problem with respect to the stacking strategies in their model.

Therefore, in the present study, we aimed to integrate the main cause of container handling, considering different stacking strategies, with the effect of different scenarios of further information about transportation mode, inflow, and outflow of the container.

In order to examine this, agent-based modeling was considered a powerful tool to model these approaches and get conclusions from the interactions between system agents. Hence, the ABM can provide a unique model for investigating the impact of further transportation mode on the container handling process in a container terminal. Therefore, the model examined the effect of the container handling process based on different scenarios. The ABM can also exhibit complex behavior patterns in a container terminal and provide valuable information about the dynamics of the real-world system that it emulates. To manage this highly interconnected network, the intelligence of agents and the average knowledge of agents [42] were taken into account. Therefore, any agents were coded to have a certain level of awareness using message communication ability when they initiate a command in the model. Moreover, in ABM, an observer can influence the dependence of knowledge spread (highly knowledgeable agents) within networks and the

way agents select other agents for knowledge acquisition by conditioning the Knowledge Management Strategy [43]. This way, we as observers can give commands to any agents to select communication with other agents after filtering for knowledge acquisition.

Accordingly, the structure of the knowledge dynamics network is determinative in improving innovation, and therefore a competitive advantage of an organization, meaning that agents with a higher knowledge level are effectively more knowledgeable than agents with a lower knowledge level [44]. This assumption is reflected in scenarios defining the impact of having information about further transportation modes on agents and terminal performance.

The ABM consisted of three major components, which are known as agents, interactions, and environment. The agent can be classified as an independent entity with specific characteristics, while each agent can behave autonomously and has the ability to sense and communicate. It should be noted that agents may have complete or incomplete information about their surroundings, and they may have the ability to impact other environments. Accordingly, the ABM model can determine instructions to a hundred or even more agents in an environment where agents can interact with each other and impact each other based on the characteristics of the defined environment.

The main goals of the present study are to provide an insight into the container handling problem concerning stacking strategy and to examine different scenarios which could give a broader view of this process. Therefore, the comprehensive technical details of the container handling process are not taken into account in the current model. The developed model can be a foundation for future and more complicated models, including more sophisticated algorithms.

The main outline of the agent-based model is presented in Figure 3. It represents a simplified interaction among different agents (terminal, sea vessels, barges, trains, and trucks).

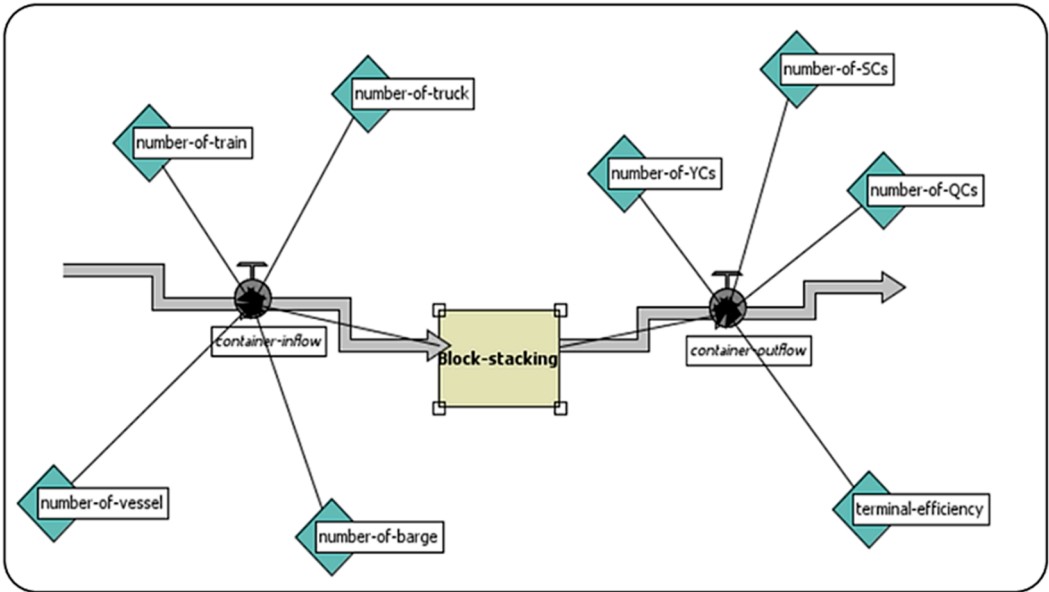

**Figure 3.** Main outline of the Agent-Based Model.

Figure 3 presents the relationship between the agents (terminal, vessel, barge, train, and truck). The container inflow is the rate that new containers enter the system, which depends on the number of container vessels, barges, trains, and trucks, and the number of containers that are already in the system. The number of the stacking-area sliders sets the number of containers in a block-stacking, which in this case was set at 300 TEU. However, this parameter is calibrated in line with the different cases. Moreover, the container outflow is dependent on various factors, such as the number and efficiency of SCs, YCs, QCs, the number of containers already on the system, and terminal efficiency. To do a sensitivity analysis and to be in line with different cases, all parameters were calibrated.

As said before, a model only represents a part of the real situation. Hence, this model only deals with the container handling process for the various inflow transportation modes, namely vessel, barge, train, and truck. However, other effective factors with minor impacts on the results of the ABM, such as detailed technical measures, container stacking rules (weight limit, dangerous goods), and how QCs, YC, and SCs are allocated, were not taken into account in the model.

Based on the ABM, container inflow was determined by the number of containers set for different modes of transport controlled by the number-of-trains (TEU) slider, number-of-trucks (TEU) slider, number-of-barges (TEU) slider, and number-of-vessels (TEU) slider. In our model, a total number of 300 TEU containers was split among Block A, Block B, Block C, and Block D. The number-of-stacking-blocks sliders controlled the number of containers in each stacking block.

In the current study, the data for scenario analysis was generated based on the ABM's output. The information was derived from a terminal operator in the Port of Antwerp. Consequently, the container terminal activity and related information with respect to the number of vessels, barges, trains, and trucks as the inflow container and the number of SCs, QCs, YCs as the outflow container were retrieved from this stakeholder. The container terminal in Port of Antwerp now contains a total of 41 quay cranes across 9 berths, with maximum depth at Chart Datum (m) 17,200 straddle carriers, and a quay length of 3700 m [45].

Thus, we tried to model scenarios that could be a reflection of real-world conditions and contain valid assumptions in line with the operational practices of the container loading/unloading process inside a terminal in the Port of Antwerp. Moreover, the validity of the process was shown by a meaningful interaction among the agents corresponding to the real-world loading/unloading process.

With respect to the SCs', QCs', and YCs' configuration, the assumptions were based on the publicly available information from the Port of Antwerp website. Configuration of SCs was as follows: 3- or 4-high stacking (one over two containers/one over three containers); lifting speed (full load), 18 m/min; lifting speed (empty), 26 m/min; driving speed (full load), 30 km/h; driving speed (empty), 30 km/h [46]. QCs were set at 41, with outreach up to 25 containers wide [47]. SCs at the terminal are responsible for both unloading/loading and stacking. Therefore, in our model, we presented different shapes named YCs but with the same characteristics as SCs. Deep-sea-going vessels are assumed to be an average of 370 m long and 55 m in width, with a capacity of 17,000 TEU, while barges' average length was 110 m and average width was 11.4 m, with a capacity of 200 TEUs [48].

Figure 4 shows the interface of the container terminal on NetLogo's agent-based modeling software version 6.2 for different stacking areas and vehicles set for each of the situations.

Considering the early developed scenarios and based on the existing equipment and berthing capacity in a container terminal of the Port of Antwerp, we assumed that three deep-sea-going vessels and one barge could berth on the quay for the (un)loading process simultaneously. Moreover, trains and trucks in the landside are also part of the (un)loading process. Every block was allocated to a QCs group as well as SCs. For instance, the Block A container loading/unloading process is only done by a group of SCs-A and QCs-A with the same technical specifications. Terminal efficiency will vary considering different performance indexes of terminal equipment for the two possible scenarios.

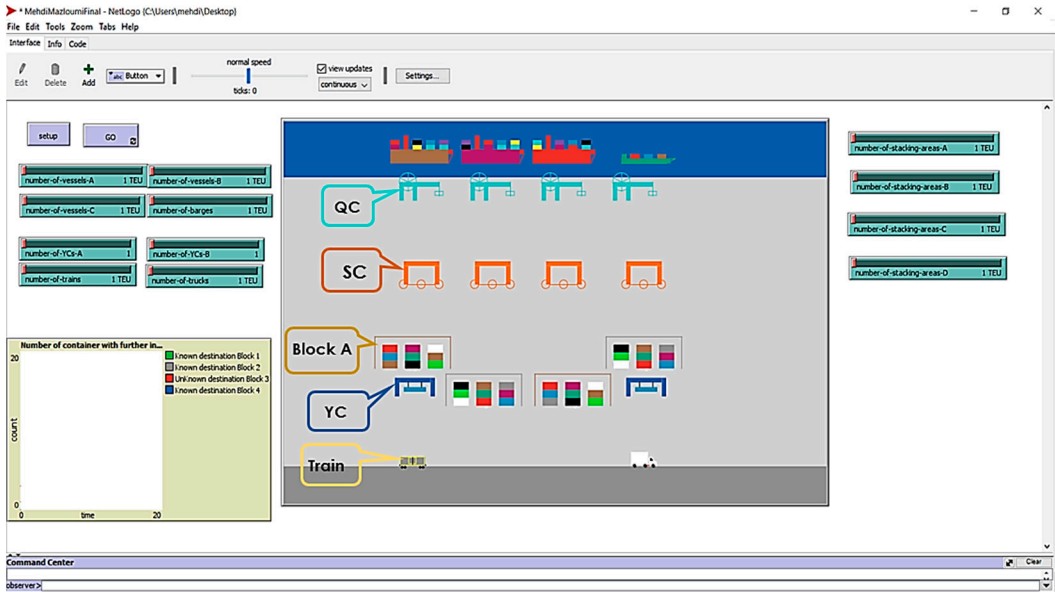

**Figure 4.** Interface of the container terminal on NetLogo agent-based modeling software.

### 3.3. Overall Equipment Effectiveness (OEE) Index

We used Overall Equipment Effectiveness (OEE) in measuring how well the operation is utilized compared to its ideal and full potential. The preferred and simplest OEE calculation is based on the OEE factors, including Availability, Performance, and Quality. Availability is determined using the ratio of Run Time to Planned Time (see Equation (1)).

$$\text{Availability} = \text{Run Time}/\text{Planned Production Time} \tag{1}$$

Run Time is calculating by Planned Production Time minus Stop Time, where Stop Time is when the process is intended to be running but unplanned stops (e.g., equipment malfunctions) or planned stops (e.g., changeovers) affect it (see Equation (2)).

$$\text{Run Time} = \text{Planned Run Time} - \text{Stop Time} \tag{2}$$

Moreover, performance is defined as the ratio of Net Run Time to Run Time (Equation (3)) and takes into account any cases that lead the current process to run at less than the maximum possible speed, including slow cycles and small stops.

$$\text{Performance} = (\text{Ideal Cycle Time} \times \text{Total Count})/\text{Run Time} \tag{3}$$

Quality in the container terminal takes into account Good Tasks, tasks that are successfully done the first time without needing any rework, and can be calculated using Equation (4):

$$\text{Quality} = \text{Good Tasks}/\text{Total Tasks} \tag{4}$$

Consequently, OEE calculates all losses, resulting in a measure of truly productive container terminal operational time, and it is calculated as Equation (5):

$$\text{OEE} = \text{Availability} \times \text{Performance} \times \text{Quality} \tag{5}$$

### 3.4. Developed Scenarios

As mentioned, two scenarios were evaluated. Each of these scenarios had different parameters affecting the model and were adjusted to determine the impact of having further information about transportation modes on each of the parameters. The parameters set for each scenario have been presented in Table 1.

**Table 1.** Parameters for scenario analysis.

| Scenarios | Parameters | Values |
|---|---|---|
| 1: Block Assignment with no further Info | Number of containers in stacking area A (with known mode of transport) | 81 TEU |
| | Number of containers in stacking area B (with known mode of transport) | 79 TEU |
| | Number of containers in stacking area C (with unknown mode of transport) | 100 TEU |
| | Number of containers in stacking area D (with known mode of transport) | 40 TEU |
| | Number of straddle carries | 4 TEU per tick |
| | Number of quay cranes | 4 TEU per tick |
| | Number of yard cranes | 2 TEU per tick |
| | Number of trains | 1 TEU per tick |
| | Number of trucks | 1 TEU per tick |
| 2: Block Assignment with further Info | Number of containers in stacking area A (with known mode of transport) | 100 TEU |
| | Number of containers in stacking area B (with known mode of transport) | 75 TEU |
| | Number of containers in stacking area C (with unknown mode of transport) | 0 TEU |
| | Number of containers in stacking area D (with known mode of transport) | 125 TEU |
| | Number of straddle carries | 4 TEU per tick |
| | Number of quay cranes | 4 TEU per tick |
| | Number of yard cranes | 2 TEU per tick |
| | Number of trains | 1 TEU per tick |
| | Number of trucks | 1 TEU per tick |

### 3.4.1. Scenario 1—Block Assignment with No Further Info

In this scenario, the study assumes that stacking area C at the terminal is dedicated to containers with no further information about transportation modes, while containers in the stacking areas A, B, and D were allocated to containers with available information about further transportation modes. This is a current situation at the terminals. There are nearly 30% to 40% of containers with unknown information about further transportation modes. This imposes an efficiency reduction at the terminal because they have to relocate and transfer containers to find the right container for the right transportation modes, whether sea transportation or land transportation.

Hence, in this case, 100 TEU containers out of 300 TEU containers have no information about the next modes of transportation. The initial numbers of SCs were set to 1 for each stacking area and 4 in total, and they were responsible for transporting containers between stacking areas (Blocks) and quaysides and the other way around. Two of the YCs were set for performing (un)loading processes in/out of the trains and trucks. Four of the QCs were set to load/unload containers in/out of the vessels and barges.

### 3.4.2. Scenario 2—Block Assignment with Further Info

This case examines what the situation of container stacking strategies and terminal efficiency would look like if further transportation mode is known. In this scenario, containers were distributed among stacking areas A, B, and D, while stacking area C was empty.

The differences between this case and the former one are that the initial numbers of SCs were set to 1 for each stacking area, and 3 in total (one SC less than the base scenario), and they were responsible for transporting containers between stacking areas (Blocks) and quaysides and the other way around. Two of the YCs were set for performing (un)loading

processes in/out of the trains and trucks. Three QCs (one less QC less than the first case) were set to load/unload containers in/out of the vessels and barges.

## 4. Results

The results of the two scenarios are presented in this section. All models ran for 744 ticks (representing 31 days × 24 h) on the NetLogo agent-based modeling software version 6.2 to evaluate the container loading/unloading process over time. OEE of the terminal, for both cases, was also calculated and is discussed in this section.

Figure 5 explains the number of containers loaded/unloaded in scenario 1.

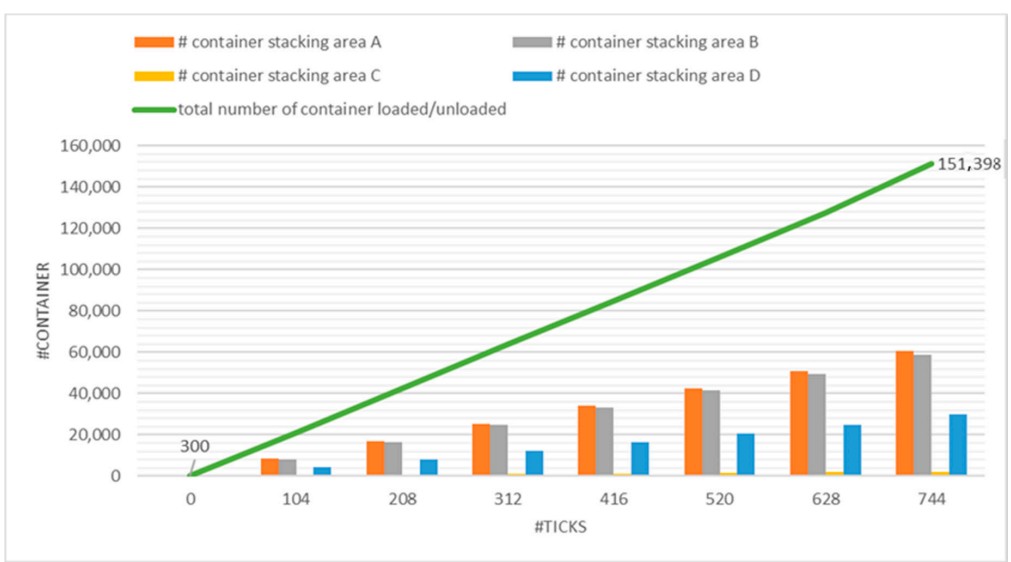

**Figure 5.** Number of containers handled in scenario 1.

As can be seen, the initial total number of containers in different stacking areas was set to be 300 TEU at time step zero. Therefore, the number of loaded/unloaded containers in stacking areas A, B, and D shows a fixed ascending trend. This situation continues throughout the time step. However, an observation of the graph suggested that the number of containers loaded/unloaded in stacking area C always remains very low, irrespective of the initial value of 100 TEU, compared to the function of other stacking areas.

This can be linked to the reality that containers with unknown destinations should be relocated or transferred to the other stacking areas to be transported via certain transportation modes. It causes a significant drop in the volume of containers being loaded/unloaded over time.

As described in the OEE factors, the calculation begins with the Planned Run Time (see Table 1). Therefore, firstly, we determined any shift time where there is nothing to stop the transportation process (typically breaks), which in our model was equal to 744 h. The next step was then to calculate the Run Time that unloading/loading was actually running. There was no stopping, including unplanned stops (equipment failure) or planned stops (set-up and adjustments). In our model, a sum of equipment failure, set up, and adjustments minus planned Run Time was equal to Run Time. Hence, the overall availability was calculated based on the result of Run Time divided by Planned Run Time.

Container terminal operators use quay crane productivity as a key indicator and one of the critical parts of overall terminal productivity at the same time. The number of moves per hour is the estimator in measuring the productivity of a QC. Almost all terminals are able to achieve maximum productivity as low as 70% and as high as 80% of the nominal performance [31]. This gap is due to productivity losses caused by operational disturbances. QCs do not achieve the technically possible productivity. Therefore, in our calculation, the ideal cycle time was assumed to be 1/400 TEU containers, which was 30% more than the

real cycle time in our model. In order to calculate overall performance, the Ideal Cycle Time was multiplied in Total Count, and the result then was divided by Run time.

The results of the OEE calculation for scenario one and related calculations have been presented in Tables 2–5.

**Table 2.** Unloading/loading information for scenario 1.

| Total Time (Month) | 744 |
|:---:|:---:|
| Not Scheduled (1 day) | 24 |
| Planned Run Time (7 days, 3 shifts) | 720 |
| Run Time | 435 |
| Stop Time | 285 |
| Ideal Cycle Time | 0.0025 |

**Table 3.** Transportation information for scenario 1.

| Transportation | Container |
|:---:|:---:|
| Total Count | 152,142 |
| Good Count | 151,950 |
| Container Reject Count | 100 |
| Startup Reject Count | 92 |
| Total Reject Count | 192 |

**Table 4.** Top losses for scenario 1.

| Top Losses | Time in Hours |
|:---:|:---:|
| **Equipment Failure (Lost Time)** | **188** |
| SCs engine failure | 48 |
| QCs software error | 10 |
| YCs engine failure | 20 |
| All Other Losses | 10 |
| QCs-C waiting time | 105 |
| **Setup and Adjustments (Lost Time)** | **92** |
| QCs initial set-up | 48 |
| SCs initial set-up | 24 |
| YCs initial set-up | 20 |
| **Performance Loss (Lost Time)** | **60** |
| **Transportation Rejects (Lost Time)** | **0** |
| **Startup Rejects (Lost Time)** | **15** |

**Table 5.** OEE calculation for scenario 1.

| OEE Analysis | OEE % |
|:---:|:---:|
| Availability | 60.42% |
| Performance | 86.44% |
| Quality | 99.87% |
| **OEE** | **52.15%** |

In Table 3, the Good Count was calculated based on the total number of containers being transported minus set-up, adjustments, and Container Reject Count. The overall quality of transportation was then determined based on the outcome of Good Count divided by Total Count.

The second scenario of loading/unloading containers with further information about transportation modes is shown in Figure 6.

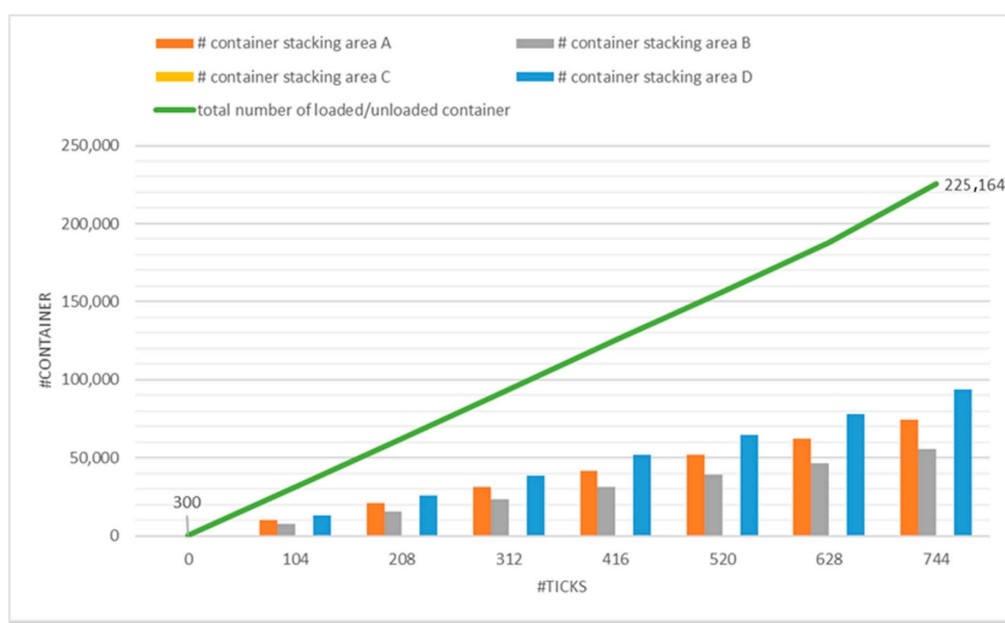

**Figure 6.** Number of containers handled in scenario 2.

In this scenario, the initial total number of containers was set the same as case 1 and equal to 300 TEU, but there are no containers in stacking area C with unknown further transportation modes. However, the graph above reveals that if all containers had information about further transportation modes, the total containers being handled would increase significantly. In this case, 73,022 TEUs are handled more than in the previous scenario.

It is eliminating extra movement and relocation minimization at the container terminal which allows a faster transportation flow. This implies that having further information about transportation modes increases the number of containers handled and creates extra space for storing more containers.

In order to calculate OEE for the second case, there was no waiting time for QCs-C. The total outflow of containers in scenario 2 increased by 73,022 TEU compared to case 1, given the fact that performance is defined as the ratio of Net Run Time to Run Time. However, in scenario 2, straddle carriers and quay cranes dedicated to stacking area C were idle. Hence, it decreased the performance rate of case 2 compared to the first scenario. The last variable is the quality of the unloading/ loading process. It has remained the same for both scenarios. The OEE index showed that the effectiveness of the container terminal increased (see Table 6), which was in line with the agent-based model's outcome and shows a direct relationship between further information for transportation modes and the OEE index increasing. A comparison of the two scenarios shows that the QCs-C waiting time was the main factor for reducing the OEE index for scenario 1, due to the ineffective function of the dedicated stacking area for this quay crane, while, in the second scenario, there was no waiting time due to a lack of information about further transportation modes.

**Table 6.** OEE calculation for scenario 2.

| OEE Analysis | OEE % |
|---|---|
| Availability | 75.00% |
| Performance | 70.44% |
| Quality | 99.91% |
| **OEE** | **52.78%** |

## 5. Discussion and Implications

Investigating the impact of further information about transportation modes on the stacking strategy and its relationship with the container terminal effectiveness brought a lot of attention to the domain of transportation research. It is also important to consider the sustainability parameter based on those indicators that have the greatest impact on the analyzed model. In order to draw more conclusions, and in the case of the sustainability calculation, the environmental impact of the container terminal is derived from the initial or final state of the analysis in each of the stacking areas, and the total environmental impact of initial state transportation in each stacking area is derived from the total initial state of the productive transportation system in the analyzed container terminal. Although different indicators such as Ecotax, Ecovalue08, and Ecoindicator-99 [49] have been proposed, $CO_2$ equivalent emissions is a commonly used indicator.

Depending on the availability of information about further transportation modes, the terminal's overall effectiveness increased from the base scenario to the alternative one. This ultimately leads to a better OEE index and more sustainability in the container terminal. However, there is concern about the level of information sharing among different stakeholders in the container transportation industry, which is the main barrier for selecting an appropriate stacking strategy at the container terminals. Small freight forwarders have claimed that big companies tend to suggest a lower price to gain their market share if they have access to their customers' detailed information. Hence, a secured data-sharing platform can be designed by the terminal operators to collect the different stakeholders' information, where each party has no fear about misuse of their data by competitors. In this sense, terminal operators play a mediator role, and they have to assure different parties that their information could reduce the total transportation cost.

This research examined the container stacking strategy using a theoretical agent-based modeling approach. It evaluated a base scenario and an alternative scenario to evaluate the impact of having more information about the next mode of transport on the terminal efficiency. Maritime transportation, particularly container transportation, forms a significant part of international trade. However, container terminals always deal with various unpredictable challenges, which is the main affecting factor for efficiency. Any container terminal needs the container inflow and outflow information to establish an effective stacking strategy, while considering other variables such as ship arrivals, cranes, SCs, AGVs, or trucks. In comparison, most container terminals are suffering from a lack of information regarding further transportation modes.

Thus, it is due to this challenge that this study aimed to propose different stacking strategy scenarios in which, in the base case, we don't have information about how the containers are handled further. Then, in an alternative scenario, we checked if and how the stacking strategy could be different if we do have this information. In achieving these objectives, first, we developed two scenarios based on the literature review for a possible solution to solve the loading/unloading problems based on different stacking strategies between the quay/yard side. Then, an agent-based model was developed to evaluate the different scenarios identified in the study. Finally, the OEE index was calculated to check the container terminal's effectiveness considering the two cases.

## 6. Conclusions and Future Research

An agent-based model approach can contribute to understanding complicated container terminal transportation systems. With the objectives identified earlier, this research provided answers to the initially identified questions. The main lines of questions were to identify the existing stacking strategy and what would happen if we had further information about transportation modes, and finally, to calculate the overall equipment effectiveness when we altered parameters based on the different scenarios.

Establishing an integrated data-sharing system has been found to be the most promising option to improve a container terminal's productivity. In this model, we show that

having information about further transportation modes can influence terminal performance and its overall equipment effectiveness.

From the modeling point of view, our model represented a part of the real situation. Therefore, this model only dealt with the container handling process for the various inflow transportation modes. On the other hand, to design a container terminal's complex, multi-disciplinary systems, designers need a design method that allows them to systematically decompose this complex design problem into simpler sub-problems. Hence, it would be interesting to investigate bay and slot stacking strategies and add more dynamics to the model regarding vessels, barges, trains, and trucks size, and AGVs. A fully automated terminal would also be an interesting case for future study cases. It should be noted that the number of containers for each stacking area was considered the same, with the average number of 300 TEU. It might also be interesting for future research to consider different numbers of containers and peak times.

Therefore, it is recommended to investigate existing obstacles for integrating container handling flow-related information. Conclusively, overall cost and related economic studies of the developed scenarios would provide a broader view of the costs and benefits of an integrated information system. Next to that, sustainability indicators such as $CO_2$ equivalent emissions could be calculated for the different cases.

All these concerns need to be addressed in future research to improve the container terminal loading/unloading process. Therefore, it is of particular interest for other scholars to proffer answers to these questions and discussions.

**Author Contributions:** M.M. performed the literature review, did the interviews, built the model, and performed the analysis. E.v.H. supervised the work. M.M. contributed by adding literature, providing input for the development of the model, and adopting OEE analysis in the paper. He also supported M.M. with the interpretation of the results, and he helped in structuring the paper. Both authors have read and agreed to the published version of the manuscript.

**Funding:** This research received no external funding.

**Institutional Review Board Statement:** Not applicable.

**Informed Consent Statement:** Not applicable.

**Data Availability Statement:** The data used in this paper was collected via publicly available data from MPET, Port of Antwerp database.

**Acknowledgments:** The authors want to thank Wijnand Visser (Manager Continuous Improvement, MPET, Port of Antwerp) for the insights he provided during the research regarding the container terminal operations in the Port of Antwerp.

**Conflicts of Interest:** The authors declare that they have no competing interest.

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
