# Peer review of "Improvement of Container Terminal Productivity with Knowledge about Future Transport Modes: A Theoretical Agent-Based Modelling Approach"

_sustainability, doi:10.3390/su13179702_

Round 1
Reviewer 1 Report
This study explores the factors impacting the container handling process considering OEE as the performance measure. The following comments should be addressed carefully before we can proceed with this manuscript.
Abstract.
- This section should be shortened to less than 180 words. The structure is sound but there are redundant and unnecessary sentences that could be integrated/removed.
- About the major findings, is increasing OEE from 52.15% to 52.78% really significant? you should present this improvement in a better way.
Introduction. This section should be completely restructured.
- Some of the information included in the first paragraph needs citation(s).
- It is not recommended to use figures in the introduction section. Instead, you could discuss the trend within the content.
- You started with a background paragraph followed by a problem statement. You should expand the section by including three additional paragraphs: (a) what has been done to address the problem (a broad perspective, i.e. multidisciplinary). (b) a critical review of the most relevant literature. (c) research gap in the literature and a clear statement of your contributions. Besides, you need to close the section by providing the outline, i.e. the rest of the manuscript.
- The logical sentences that justify the importance of your study, like those in Lines 201-205 should be mentioned in the new introduction section. Do not forget that the introduction section should encourage the potential readers to go through the rest of your manuscript.
Literature Review. This section should be improved by applying a story-telling manner in your review of the literature. Following a broad-to-narrow flow of information, there must be a logical connection between the paragraphs. I would like to emphasize that the literature review section should be deeper compared to the paragraph presented in the literature review. The objective is to exhaust ALL the relevant articles, and not only the most relevant onces. Overall, it should be written in a way that the identified gap in the introduction section is well supported.
Method.
- Section 3, the research approach, should be included as an introductory sub-section to the methodology section.
- You mentioned that "scenarios were developed based on evidence from the literature"; you should elaborate on and provide citations.
- Why OEE is used? you need to add enough justification with citations.
- What is ABM? you cannot use abbreviations before defining them (you defined it later in Line 184).
- What are the similar application areas of ABM? how would you justify the use of ABM for this context? you need to clarify in the methodology section.
- The authors are highly recommended to provide a step-by-step explanation of the methodology to make it replicable for the potential readers.
- The contents provided in section 4.1 should be well organized. In its current form, it is misleading and confusing.
- You need to provide justification for the assumptions starting from Line 283.
- Section 4.3.2 should be Scenario 2!
- Your justifications for the selected scenarios are not convincing. Why didn't you consider additional scenarios? considering only two situations is too simplistic and cannot reflect where exactly is the trigger point of the resulting improvement.
Results. You mentioned that "The data used in this paper was collected via publicly available data from MPET, port of Antwerp database". However, this is not enough and you need to provide all the details including the data collection process and the data used in establishing your model.
Discussion and implications. You submitted your work to the Sustainability journal yet you dismissed to provide implications of your work for sustainability.
Conclusions. This section should be restructured to include the following information. Paragraph (1): background, your contributions. Paragraph (2): major findings and implications. Paragraph (3): limitations of this study and directions for future research works.
Overall, I am not convinced that your research design, i.e. the selection of the scenarios, is sound. This shortcoming also impacts the results analysis section, making it rather weak. Besides, the presentation of this work needs substantial improvement.
Author Response
We would deeply thank you for your very good and useful comments of your respected reviewers on our manuscript entitled: Improvement of container terminal productivity with knowledge about future transport modes: A theoretical Agent-Based Modelling Approach “. we made changes in the structure and content of the article in line with your comments. We wish these corrections or modifications could fulfil the requirements of the respected reviewers.

Reviewer 2 Report
The paper studies the problem of limited information about further transportation modes of a container, leading to extra movements and relocations of the container, inside the stacking area. The authors adopt the Overall Equipment Effectiveness Index (OEE) in order to evaluate the container terminal's effectiveness. Results showed that having information on further transportation mode significantly increased the container outflow, and the overall OEE increased from 52.15 % to 52.78%. The paper is interesting and the results well-presented. Also, the agent-based modeling approach is the most appropriate to perform “what-if scenario analysis”. The significant role of "available information", resulting in a higher effectiveness of the system as a whole, is universal in complex systems. Therefore, should be emphasized. This is related to the concept of "bounded rationality" for decision making. A key characteristic of "rational intelligent agents", making decisions under limited information-awareness. As a result, authors should include the previous work of the following 3 relevant papers, making also "a link" with their work:
https://doi.org/10.3390/math9010103
https://doi.org/10.1016/j.physa.2018.06.003
https://doi.org/10.1016/j.physa.2017.09.078
Author Response
The ABM can exhibit complex behavior patterns in a container terminal and provide valuable information about the dynamics of the real-world system that it emulates. To manage this highly interconnected network, the intelligence of agents and the average knowledge of agents (Ioannidis et al., 2021) were taken into account. Therefore, any agents were coded to have a certain level of awareness using message communication ability when they initiate a command in the model. Moreover, in ABM, an observer can influence the dependence of knowledge spread (highly knowledgeable agents) within networks and the way agents select other agents for knowledge acquisition by conditioning Knowledge Management Strategy (Ioannidis et al., 2018b). So that, we as an observer can give commands to any agents to select communicating other agents after filtering for knowledge acquisition.
Accordingly, the structure of the knowledge dynamics network is determinative in improving innovation, and therefore a competitive advantage of an organization. Meaning that agents with knowledge level are effectively more knowledgeable than agents with a lower knowledge level (Ioannidis et al., 2018a). This assumption reflected in scenarios defining that what is the impact of having information about further transportation modes on agents and terminal performance.

Round 2
Reviewer 1 Report
There are two minor issues that should be addressed before we can proceed with your manuscript:
- You provided a detailed literature review to support a research gap. However, you did not review the MOST relevant works that have addressed the stated problem in the introduction. Please make sure to add a paragraph containing a critical review of the most relevant works based on which you should HIGHLIGHT the research gap. You can then present your research questions after stating your research contribution(s).
- There are many limitations to simulation modeling. Please list the ones that have impacted your study the most in the conclusions section. On this basis, you may also offer additional suggestions for future research.
- The discussions and implication section should be extended. Your contribution is rather practical; a practical contribution should offer managerial insights for the private and public sectors. For example, the discussions on the sustainability aspect can be extended considering the triple bottom line. The role of sustainability innovation is another worthwhile topic to discuss. As a final suggestion, authors could offer academic insights based on the findings, e.g. how your findings may have implications for future research on the optimization of multimodal operations.
Author Response
We would deeply thank you for your very good and useful comments of your respected reviewers on our manuscript entitled: Improvement of container terminal productivity with knowledge about future transport modes: A theoretical Agent-Based Modelling Approach “. we made changes in the structure and content of the article in line with your comments. We wish these corrections or modifications could fulfill the requirements of the respected reviewers.
Reviewer 1:
Comment 1:
Reviewer comment: You provided a detailed literature review to support a research gap. However, you did not review the MOST relevant works that have addressed the stated problem in the introduction. Please make sure to add a paragraph containing a critical review of the most relevant works based on which you should HIGHLIGHT the research gap. You can then present your research questions after stating your research contribution(s).
Response: In container transportation literature, the adverse effects of demand uncertainty or disturbances in services are commonly acknowledgeable (Steadieseifi et al., 2014; Gumuskaya et al., 2020; van der Horst et al., 2019). Specifically for container terminals stacking planning, the low reliability of deep-sea vessels, having no information about further transportation modes and equipment preparation are major issues in operational level planning. In addition to the uncertainty in container arrivals, dynamism complicates container terminal planning. A terminal operator has to decide which QCs/YCs to load/unload containers, and which stacking areas are dedicated to the inbound/outbound containers. During this time gap, other information impact terminal operator decision making such as new coming orders, delays, and cancellations (Pillac et al., 2013). Uncertainty and dynamism lead to a situation in which the efficiency of operational plans and the competitiveness of container terminals decrease. This is addressed in the manuscript.
2:
Reviewer comment: There are many limitations to simulation modeling. Please list the ones that have impacted your study the most in the conclusions section. On this basis, you may also offer additional suggestions for future research.
Response: From the modeling point of view, our model represented a part of the real situation. Therefore, this model only dealt with the container handling process for the various inflow transportation modes. On the other hand, to design container terminal’s complex, multi-disciplinary systems, designers need a design method that allows them to systematically decompose this complex design problem into simpler sub-problems. Hence, it would be interesting to investigate bay and slot stacking strategies and add more dynamics to the model regarding vessels, barges, trains and trucks size, and AGVs. A fully automated terminal also would be an interesting case for future study cases. It should be noted that the number of containers for each stacking area considered being the same, and the average number of 300 TEU. It might also be interesting for future research to consider the different number of containers and peak time.
Next to that, sustainability indicators such as CO2 equivalent emission would be calculated for the different cases. This is addressed in the manuscript.
3:
Reviewer comment: The discussions and implication section should be extended. Your contribution is rather practical; a practical contribution should offer managerial insights for the private and public sectors. For example, the discussions on the sustainability aspect can be extended considering the triple bottom line. The role of sustainability innovation is another worthwhile topic to discuss. As a final suggestion, authors could offer academic insights based on the findings, e.g. how your findings may have implications for future research on the optimization of multimodal operations.
Response: Investigating the impact of further information about transportation modes on the stacking strategy and its relationship with the container terminal effectiveness derived a lot of attention on the transportation domain researches. It is also important to consider sustainability parameters based on those indicators that have the greatest impact on the analyzed model. In order to draw more conclusions, and in the case of the sustainability calculation, the environmental impact of the container terminal is referred to the initial or final state of the analysis in each of the stacking areas, and the total environmental impact of initial state transportation in each stacking area is referred to the total initial state of the productive transportation system in a container terminal analyzed. Although different indicators such as Ecotax, Ecovalue08, Ecoindicator-99 (Ahlroth & Finnveden, 2011) have been proposed, the CO2 equivalent emission is a commonly used indicator.
A secured data-sharing platform can be designed by the terminal operators to collect the different stakeholder's information where each party had no fear about misusing their data by competitors. In this sense, terminal operators play a mediator role, and they had to assure different parties that their information could reduce the total transportation cost. This is addressed in the manuscript.
